# Incidence of Cutaneous Immune-Related Adverse Events and Outcomes in Immune Checkpoint Inhibitor-Containing Regimens: A Systematic Review and Meta-Analysis

**DOI:** 10.3390/cancers16020340

**Published:** 2024-01-13

**Authors:** Nina B. Curkovic, Kun Bai, Fei Ye, Douglas B. Johnson

**Affiliations:** 1School of Medicine, Vanderbilt University, Nashville, TN 37232, USA; 2Vanderbilt Ingram Cancer Center, Department of Biostatistics, Vanderbilt University Medical Center, Nashville, TN 37232, USA; 3Vanderbilt Ingram Cancer Center, Department of Medicine, Vanderbilt University Medical Center, Nashville, TN 37232, USA; douglas.b.johnson@vumc.org

**Keywords:** immune checkpoint inhibitors, cutaneous immune-related adverse events, autoimmune toxicities, anti-angiogenic, chemotherapy, melanoma, renal cell carcinoma, non-small cell lung cancer, urothelial carcinoma, meta-analysis

## Abstract

**Simple Summary:**

Immune checkpoint inhibitors are increasingly being used in the treatment of a variety of cancers, both alone and in combination with other cancer therapies. Side effects often include skin reactions, which may occur more frequently in therapeutic regimens consisting of multiple immune checkpoint inhibitors. We conducted a systematic review and meta-analysis of clinical trials to better understand the frequency of skin reactions secondary to immune checkpoint inhibitors, known as cutaneous immune-related adverse events, across several different immune checkpoint inhibitor regimens, doses, and cancers. Our analysis provides benchmark incidence rates for cutaneous immune-related adverse events including pruritis, rash, and vitiligo, and validates previously reported links between the development of cutaneous immune-related adverse events and outcomes of therapy.

**Abstract:**

Immune checkpoint inhibitors (ICIs) are used to treat many cancers, and cutaneous immune-related adverse events (cirAEs) are among the most frequently encountered toxic effects. Understanding the incidence and prognostic associations of cirAEs is of importance as their uses in different settings, combinations, and tumor types expand. To evaluate the incidence of cirAEs and their association with outcome measures across a variety of ICI regimens and cancers, we performed a systematic review and meta-analysis of published trials of anti–programmed death-1/ligand-1 (PD-1/PD-L1) and anti–cytotoxic T lymphocyte antigen-4 (CTLA-4) ICIs, both alone and in combination with chemotherapy, antiangiogenic agents, or other ICIs in patients with melanoma, renal cell carcinoma, non-small cell lung cancer, and urothelial carcinoma. Key findings of our study include variable cirAE incidence among tumors and ICI regimens, positive association with increased cirAE incidence and response rate, as well as significant association between increased vitiligo incidence and overall survival. Across 174 studies, rash, pruritis, and vitiligo were the most reported cirAEs, with incidences of 16.7%, 18.0%, and 6.6%, respectively. Higher incidence of cirAEs was associated with ICI combination regimens and with CTLA-4-containing regimens, particularly with higher doses of ipilimumab, as compared to PD-1/L1 monotherapies. Outcome measures including response rate and progression-free survival were positively correlated with incidence of cirAEs. The response rate and incidence of pruritis, vitiligo, and rash were associated with expected rises in incidence of 0.17% (*p* = 0.0238), 0.40% (*p* = 0.0010), and 0.18% (*p* = 0.0413), respectively. Overall survival was positively correlated with the incidence of pruritis, vitiligo, and rash; this association was significant for vitiligo (*p* = 0.0483). Our analysis provides benchmark incidence rates for cirAEs and links cirAEs with favorable treatment outcomes at a study level across diverse solid tumors and multiple ICI regimens.

## 1. Introduction

Immune checkpoint inhibitors (ICIs) have proven to be effective therapies for a variety of cancer types. By targeting immune checkpoints with anti-PD-1, anti-PD-L1, anti-CTLA4, and, more recently, anti-LAG-3 agents, ICIs disinhibit the immune system to unleash anti-tumor immune responses. However, clinically significant off-target effects due to the generation of autoreactive T cells, termed immune-related adverse events (irAEs), are common and can affect virtually any organ [1]. Cutaneous immune-related adverse events (cirAES) are among the most frequently encountered irAEs, occurring at rates of 30–40% with anti-PD-1 monotherapy, with potentially higher incidence and severity with combination regimens [1,2]. The clinical presentation of cirAEs is variable and may range from common eczematous and lichenoid eruptions to, less commonly, bullous pemphigoid and reports of associated cutaneous infections [3].

Several retrospective studies have suggested that cirAEs are associated with improved responses and survival [4,5,6,7]. These associations may be explained by shared antigens between the skin and tumors; a study on non-small cell lung cancer (NSCLC) patients treated with anti-PD-1 therapy identified T cell antigens that were shared between tumor tissue and skin, highlighting a mechanism by which the development of cirAEs could be associated with therapeutic benefit [8]. ICI-induced vitiligo, a cirAE more commonly seen in patients with melanoma, has been suggested to occur via ICI-induced loss of immune privilege to normal melanocytes following the release of shared melanocytic antigens within destroyed melanoma tumor cells [9].

Importantly, therapeutic benefits may also arise from other patient and tumor-specific factors. PDL-1 expression, metastatic site, tumor mutational burden, tumor-infiltrating lymphocytes, and the gut microbiome have been implicated in influencing the efficacy of ICIs [10,11].The development of cirAEs may be just one aspect of developing more individualized treatment prognostications as more information regarding patient-specific factors becomes available for use in clinical practice.

Prior meta-analyses have predominantly examined cirAEs in association with ICI monotherapy across cancer types or combined ICI regimens in a single cancer type [12,13,14]. As ICIs are increasingly being used in combination with multiple distinct immune and non-immune-based regimens, the incidence rates of various types of cirAEs across distinct classes of ICI-based therapies need to be elucidated. In this study, we address cirAE incidence in these novel ICI-based regimens and further examine cirAEs among several tumor types to broaden what is known about these common adverse effects. Further, the impact of cirAE incidence on survival, response, and duration of therapy has not been robustly examined across ICI-based regimens. In this meta-analysis, at the study level, we examine the incidence and prognostic associations of cirAEs across solid tumors and multiple ICI regimens, including monotherapy and combination with anti-angiogenic agents, chemotherapy, or other ICIs.

## 2. Materials and Methods

### 2.1. Study Identification

We searched PubMed, Embase, and ClinicalTrials.gov (accessed on 15 October 2023) to identify clinical trials with immune checkpoint inhibitors to include in this analysis.

PubMed was searched for clinical trials using the following terms: “atezolizumab”, “avelumab”, “cemiplimab”, “dostarlimab”, “durvalumab”, “ipilimumab”, “nivolumab”, “pembrolizumab”, “melanoma”, “carcinoma, transitional cell”, “bladder cancer”, “urothelial carcinoma”, “carcinoma, non-small-cell lung”, “non small lung carcinoma”, “non small lung cancer”, “carcinoma, renal cell” and “renal cell carcinoma.” Results were filtered using the criteria of human studies in the English language to identify 1848 studies that spanned from 2005 to 2022. Similar search terms were used for ClinicalTrials.gov (yielding 281 results) and Embase (yielding 2471 results).

All queries were performed on 4th April 2022. Duplicate studies were removed prior to the screening of a total of 3860 studies for inclusion or exclusion utilizing Covidence systematic review software (Veritas Health Innovation, Melbourne, Australia. Available at www.covidence.org, accessed on 15 October 2023).

### 2.2. Study Selection

Studies were included if they (1) reported incidence of cutaneous treatment-related and/or irAEs; (2) enrolled at least 30 patients and reported cirAEs within at least one study cohort which enrolled a minimum of 30 patients; (3) were prospective clinical trials in adult patients with either melanoma, renal cell carcinoma, urothelial carcinoma, or non-small cell lung cancer; and (4) involved ICIs given intravenously either alone or in combination with another ICI, chemotherapy, or antiangiogenic therapies (including bevacizumab, lenvatinib, axitinib, sunitinib, or cabozantinib). We included only the above cancer types to allow for comparison across cancer types.

Studies were excluded if they (1) involved ICIs in combination with radiotherapy or an agent not listed in the inclusion criteria; (2) involved neoadjuvant ICIs; (3) reported pooled data from multiple trials; or (4) reported on a study subgroup and had the entire trial data reported elsewhere. When multiple publications reported on the same trial, the manuscript with the longer follow-up time was selected unless cirAEs were not reported, in which case the article with the next longest available follow-up with data on cirAEs was selected.

A total of 396 studies were included in the full-text review following screening of the initial title and abstract. Following full-text review, 174 studies were included (Figure 1). Two reviewers reviewed studies during title and abstract screening, and one reviewer extracted data from the included studies. A second reviewer was available to review select full-text articles to determine the additional exclusion of studies during the full-text review. The systematic review followed the recommendations of the Preferred Reporting Items for Systematic Reviews and Meta-Analyses (PRISMA). The protocol has not been registered.

### 2.3. Data Extraction

The data collected included trial phase, tumor type, treatment, treatment class (e.g., anti-PD-1 or anti-VEGF), dose of anti-CTLA-4 agents (in mg/kg) when applicable, ICI treatment duration (median months), objective response rate (ORR), duration of response (DOR, median months), overall survival (OS, median months), progression-free survival (PFS, median months), and follow-up length (median months) when available. If the duration of treatment was recorded according to doses of the treatment, the duration of ICI treatment was estimated in months utilizing the reported dose frequency and schedule. In studies including patients with brain metastases and reporting both intracranial and extracranial ORR, intracranial ORR was collected.

Data on cirAEs were recorded according to treatment-related adverse events within the text or Appendix A. The overall number of patients with an unspecified cirAE (all grade, grade 3–4, and grade 5) was collected when available. Additionally, commonly reported categories of cirAEs, including pruritis, vitiligo, rash, and maculopapular rash (all grade and grade 3–4), were recorded when available. Other specific cirAEs were noted, such as erythema multiforme, pemphigoid, psoriasis, severe cutaneous reaction, or stomatitis, but were not formally analyzed given their low numbers.

### 2.4. Statistical Methods

We estimated the overall incidence of adverse effects across studies, including grouped cirAEs, rash, pruritis, and vitiligo, using meta-analysis. The random-effects model was employed to provide an estimate of the overall effect while accounting for potential between-study variability (heterogeneity was assessed using Cochran’s Q test and I^2^ statistic) [15,16]. The incidence of cirAEs by subgroup of cancer type, class of drug, and dose of ipilimumab was evaluated via random-effects models for grouped cirAEs, rash, and pruritis. Due to the smaller sample sizes of studies reporting vitiligo, which were predominantly studies on melanoma given that vitiligo is a cirAE more commonly associated with melanoma, a fixed-effect model was utilized for estimates of vitiligo incidence. Associations between cirAEs and outcome measures of ORR, OS, PFS, and DOR were estimated in separate meta regression models, adjusted according to the phase of the trial (early: phase 1/2 vs. late: phase 3/4), treatment type, and tumor type. A *p*-value of <0.05 was considered statistically significant. 95% confidence intervals were provided for all point estimates. All statistical analyses were performed using R version 4.2.1. Due to the nature of the prediction model, some of the predicted values fell outside the [0, 1] range. The corresponding bubble plot figures have been cropped to exclude these unrealistic values.

## 3. Results

### 3.1. Overall Incidence of cirAEs among All Studies

A total of 396 studies were retrieved for full-text review following initial screening of the title and abstract. Based on inclusion criteria, 174 studies were included, comprising 219 cohorts with a total of 46,134 patients [17,18,19,20,21,22,23,24,25,26,27,28,29,30,31,32,33,34,35,36,37,38,39,40,41,42,43,44,45,46,47,48,49,50,51,52,53,54,55,56,57,58,59,60,61,62,63,64,65,66,67,68,69,70,71,72,73,74,75,76,77,78,79,80,81,82,83,84,85,86,87,88,89,90,91,92,93,94,95,96,97,98,99,100,101,102,103,104,105,106,107,108,109,110,111,112,113,114,115,116,117,118,119,120,121,122,123,124,125,126,127,128,129,130,131,132,133,134,135,136,137,138,139,140,141,142,143,144,145,146,147,148,149,150,151,152,153,154,155,156,157,158,159,160,161,162,163,164,165,166,167,168,169,170,171,172,173,174,175,176,177,178,179,180,181,182,183,184,185,186,187,188,189,190]. We examined the incidence of pruritis, vitiligo, and rash across all studies reporting these cirAEs. In total, 131 studies (178 cohorts, 38,736 patients) focused on pruritis; the incidence was estimated as 18.0% (95% CI 16.4–19.7) by random-effects modeling (Appendix A). The estimated incidences of vitiligo and rash from 31 studies, largely studies on melanoma reporting on vitiligo (40 cohorts, 7693 patients), and 142 studies reporting on rash (190 cohorts, 42,332 patients) were 6.6% (95% CI 6.0–7.2) and 16.7% (95% CI 15.1–18.4), respectively (Appendix A). We also evaluated the incidence of cirAEs across all studies reporting grouped cirAEs, comprising 65 cohorts from 45 studies and 15,850 patients. The overall incidence of all-grade cirAEs was 34.8% (95% CI 30.6–39.1), as estimated by random-effects modeling (Appendix A).

### 3.2. Overall Incidence of cirAEs by Drug Class

Since the incidence of cirAEs likely varies by drug class, we grouped studies into the following regimens: PD-1 monotherapy; PD-L1 monotherapy; immunotherapy (IO) combination therapy (CTLA-4 in combination with either PD-1 or PD-L1 agents); IO + antiangiogenic agents; IO + chemotherapy (chemotherapy with any ICI agent with or without an antiangiogenic drug); and CTLA-4 monotherapy.

In random-effects modeling of studies reporting rash (142 studies), the highest incidence of rash was seen for the IO combination (25.1%), followed by the IO + antiangiogenic combination (24.9%), CTLA-4 monotherapy (24.8%), and IO + chemotherapy (22.3%). Rash was less frequent in anti-PD-1 (11.0%) and PD-L1 monotherapy (7.9%) (Figure 2). Similarly, the incidence of pruritis (reported by 131 studies) was highest in the IO combination group at 28.6%, followed by CTLA-4 monotherapy (25.2%), IO + chemotherapy (20.1%), IO + antiangiogenic (16.1%), PD-1 monotherapy (14.5%) and PD-L1 monotherapy (8.8%) (Figure 3). There were statistically significant differences between subgroups for both rash and pruritis (*p* < 0.001). A fixed-effect model was used to examine incidence of vitiligo between drug subgroups due to the smaller sample size per grouping (31 studies total). No studies on IO + antiangiogenic combination regimens reported vitiligo, possibly due to the infrequent use of these agents in melanoma cases, where vitiligo is more commonly encountered. The incidence of vitiligo, as estimated by the fixed-effect model, was highest for the IO combination (10.1%) and PD-1 monotherapy (7.9%), followed by the IO + chemotherapy (3.7%), CTLA-4 monotherapy (3.2%), and PD-L1 monotherapy (0.54%) groups (*p* < 0.001) (Figure 4).

Through random-effects modeling, the highest incidence of all-grade cirAEs, analyzed in studies reporting cirAEs, was seen in the IO combination group at 48.9% and IO + chemotherapy at 44.8%, although a single IO + anti-angiogenic cohort showed very high rates of cirAEs (78.8%). The overall incidence of all-grade cirAEs for ICI monotherapy regimens was higher for CTLA-4 monotherapy (40.6%) compared with PD-1 (22.4%) or PD-L1 (16.9%) monotherapy (Appendix A). Differences between subgroups were statistically significant (*p* < 0.0001).

### 3.3. Overall Incidence of cirAEs by Cancer Type

The incidences of rash, pruritis, and vitiligo were highest in melanoma cases. The incidence of rash was 12.4% for NSCLC, 10.5% for urothelial carcinoma, 25.3% for melanoma, and 17.7% for RCC (*p* < 0.001). The incidence of pruritis was 10.4% for NSCLC, 14.9% for urothelial carcinoma, 28.2% for melanoma, and 16.8% for RCC (*p* < 0.001). The incidence of vitiligo by cancer type was assessed via a fixed-effect model, as very few studies outside of those on melanoma reported vitiligo. The incidence of vitiligo was 7.9% in melanoma and ranged from 0.2 to 3% in the remaining cancer groups (*p* < 0.001).

Melanoma, urothelial carcinoma, and renal cell carcinoma were found to have similar incidences of all-grade cirAEs by random-effects modeling for studies reporting grouped cirAEs: 40.1%, 42.1%, and 41.3%, respectively. All-grade cirAEs were lower in non-small-cell lung cancer, at 20.8% (*p* < 0.001 for between-group differences).

### 3.4. cirAEs and Dose of CTLA-4 Regimens

To determine whether anti-CTLA-4-associated cirAEs were dose-related, we assessed the incidence of adverse events according to the ipilimumab monotherapy dose (1 mg/kg and under, 3 mg/kg, and 10 mg/kg) and the ipilimumab dose in combination with anti-PD-1 agents (1 mg/kg and 3 mg/kg).

The incidence of pruritis was highest in the ipilimumab 3 mg/kg + anti-PD-1 group at 34.8%, followed by the ipilimumab 10 mg/kg (32.9%), ipilimumab 1 mg/kg + anti-PD-1 (25.9%), ipilimumab 3 mg/kg (23.0%), and ipilimumab 1 mg/kg or under groups (2.8%) (Appendix A). Similarly, the incidence of rash was highest in the ipilimumab 10 mg/kg group at 35.5%, followed by the ipilimumab 3 mg/kg + anti-PD-1 (30.5%), ipilimumab 1 mg/kg + anti-PD-1 (21.7%), ipilimumab 3 mg/kg (21.3%), and ipilimumab 1 mg/kg or under groups (4.2%) (Appendix A). Studies reporting vitiligo only encompassed three dosing regimens; the incidence of vitiligo was highest for ipilimumab 1 mg/kg + anti-PD-1 at 14.7%, followed by the ipilimumab 3 mg/kg + anti-PD-1 (8.5%) and ipilimumab 3 mg/kg (3.2%) groups. Tests for differences between subgroups for each adverse event type were significant (*p* < 0.001) (Appendix A). The incidence of overall cirAEs by random-effects modeling was highest in the ipilimumab 3 mg/kg + anti-PD-1 group at 52.6%, followed by ipilimumab 10 mg/kg (48.5%), ipilimumab 1 mg/kg + anti-PD-1 (45.7%), ipilimumab 3 mg/kg (37.4%), and ipilimumab 1 mg/kg or under (12.5%) (Appendix A). Overall, cirAE incidence appeared to be dose-related when ipilimumab was used as a monotherapy, with less clear trends in the combination regimens.

### 3.5. Treatment Duration and cirAEs

To assess whether cirAEs were associated with the duration of treatment, we fitted meta-regression models to adverse event incidence with median treatment duration, adjusting for tumor type, treatment class, and phase of trial. While the median treatment duration was not found to be associated with pruritus or vitiligo, it was positively associated with rash (*p* = 0.0116) (Figure 5 and Appendix A). Additionally, the later phases of the trials on melanoma (compared to NSCLC, RCC, or urothelial carcinoma) were associated with increased cirAE incidence, while the treatment class (specifically PD-1 and PD-L1 blockade) was associated with lower cirAE incidence.

### 3.6. Outcome Measures and cirAEs; Response Rate, Duration of Response, Progression-Free Survival, and Overall Survival

Given the previously published link between cutaneous and other irAEs and improved outcomes at an individual patient level, we performed a meta-regression analysis of adverse event incidence for rash, pruritis, and vitiligo, including clinical outcome measures. Covariates included trial phase, drug class, and tumor type. The response rate was positively correlated with the incidence of pruritis (*p* = 0.0238), vitiligo (*p* = 0.0010), and rash (*p* = 0.0413) (Figure 6). The duration of the response was positively correlated with the incidence of rash, pruritis, and vitiligo, although these associations were not statistically significant (Appendix A). Overall survival was positively associated with the incidence of vitiligo (*p* = 0.0483) (Figure 7). Progression-free survival was positively associated with the incidence of pruritis (*p* = 0.0207) and rash (*p* = 0.0351), whereas a negative association was found with vitiligo (*p* = 0.0029) (Appendix A).

## 4. Discussion

In our analysis, we provide the most comprehensive meta-analysis of cirAEs by type, therapy regimen, and tumor, and we correlate cirAEs with outcomes across clinical trials. Across studies, cirAEs were more frequent with regimens containing combinations of ICIs and in melanoma compared to other tumor types, highlighting the impact of new and more frequently used combination regimens, such as that of ICI and anti-angiogenic drugs or dual checkpoint blockade with CTLA-4 agents. We observed general associations between the incidence of cirAEs and improved treatment outcomes, as well as with treatment duration, demonstrating the potential impact of cirAE development as a prognostic tool for mapping therapeutic outcomes and the risk of late-onset cirAEs.

Our analysis provides a benchmark for the incidence of cirAEs with distinct ICI-based therapeutic classes, and demonstrates the variability of cirAEs between regimens. Treatment regimens with combined ICIs, as well as CTLA-4-containing regimens, were consistently associated with higher incidence of all cirAEs analyzed, while PD-1/PD-L1 monotherapy regimens were associated with lower incidence. ICIs in combination with antiangiogenic agents were associated with higher rates of rash than pruritis, while no studies using this regimen reported vitiligo. This may reflect mechanistic differences by which varying agents induce differing cirAEs, and in some cases, differences in the settings in which new regimens are used. For example, combination PD-1/CTLA-4 blockade may appear to produce more vitiligo, but this may be due to its more frequent use in melanoma, whereas anti-angiogenic agents are not used in melanoma. On the other hand, cirAEs may appear more frequently in one tumor type (e.g., melanoma) due to the more frequent use (and at higher doses) of CTLA-4 blockade in that tumor type.

This analysis also highlights the association between cirAE types and therapeutic outcomes not previously reported by other meta-analyses [12,13,14]. A prior meta-analysis examining 15 studies of ICI monotherapy did not find significant associations between rash, pruritis, and overall survival or progression free survival, although it did not include vitiligo or combination ICI regimens in the study [12]. In our analysis, rash, pruritis, and vitiligo were significantly associated with response rate and PFS, as was vitiligo and overall survival. Despite observing significant associations between vitiligo and improved response rate and overall survival, we found vitiligo to be negatively associated with PFS, which may be due to the small sample size and 1–2 outlier studies. All other outcome measures, including response rate, duration of response, and overall survival, demonstrated generally positive correlations with each type of cirAE. Rash also was associated with the duration of therapy, highlighting the risk of later-onset cirAEs [191,192].

Limitations to our analysis are important to consider. Subtypes of cirAEs were reported at various levels of detail across studies, with determination of whether more specific adverse events were to be reported as part of broad groupings (e.g., rash and pruritis) left up to the investigators in each study. This may contribute to significant variability between studies and, thus, the accuracy of the results of this study. While most studies reported adverse events based upon the CTCAE, no criteria specific to more nuanced cirAE reporting exists, and the involvement of dermatologists in the reporting of cirAE likely varied across studies. Furthermore, we examined all data at the study level and not at the individual patient level, which also constitutes a source of bias in this analysis.

While meta-regression can offer valuable insights into the relationship between study characteristics and effect sizes in meta-analysis, due to the aggregated nature of meta-analysis, associations detected at the study level may not accurately reflect individual-level relationships. The interpretation of findings from study-level associations for individual studies should be carried out cautiously. Additionally, meta-regression assumes linearity in the relationship between study characteristics and effect sizes. However, this assumption may not hold in all cases, and the relationship could be nonlinear or exhibit threshold effects. In our analysis, we opted to use a linear model like meta-regression so as not to reduce the power of the analysis. Nonetheless, the assumption of linearity is an important consideration when interpreting the results of this meta-analysis.

Our analysis validates the associations of cirAEs with clinical outcomes at a study level across a range of solid tumors and ICI monotherapy and combination regimens, and provides benchmark incidence rates across these regimens. As the use of ICIs in a variety of therapeutic combinations and tumors increases, future studies capturing more detail regarding specific cirAEs and patient-level data may allow for more specific prognostic associations to be drawn and inform the care of individual patients. Ultimately, further understanding of the mechanisms by which cirAEs influence therapeutic response may assist oncologists and dermatologists in the management of both diseases and adverse events, as well as patient-specific counseling regarding prognosis.

## 5. Conclusions

Our analysis provides a benchmark for the incidence of cirAEs with distinct ICI-based therapeutic classes. It demonstrates the variability of cirAE incidence between regimens, as well as the cirAE type and associated therapeutic benefit. We found that treatment regimens with combined ICIs, as well as CTLA-4-containing regimens, were consistently associated with higher incidence of all cirAEs analyzed, while PD-1/PD-L1 monotherapy regimens were associated with lower incidence. In our analysis, response rate and PFS were significantly associated with increased rash, pruritis, and vitiligo incidence. Positive correlations were seen between increased cirAE incidence of all types, duration of response, and overall survival, though this was significant only for vitiligo and overall survival.

## Figures and Tables

**Figure 1 cancers-16-00340-f001:**
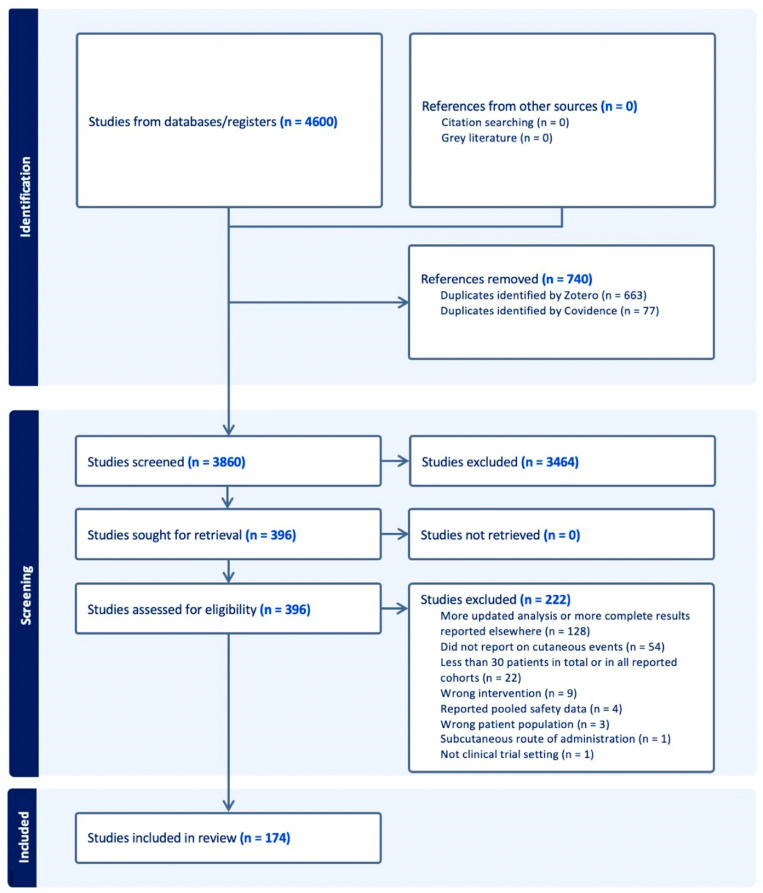
PRISMA flowchart of included studies.

**Figure 2 cancers-16-00340-f002:**
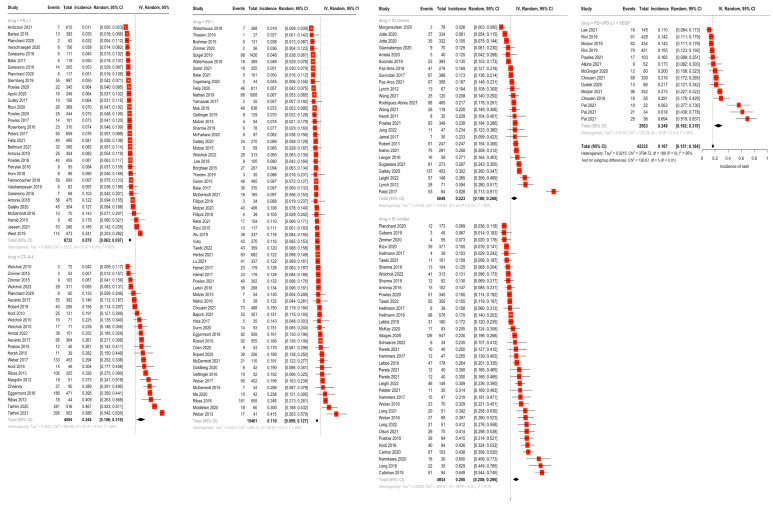
Overall incidence of rash by drug class groupings via random-effects modeling of 142 studies reporting rash. Highest incidence of rash was seen for IO combination (25.1%) followed by IO + antiangiogenic combination (24.9%), CTLA-4 monotherapy (24.8%), and IO + chemotherapy (22.3%). Rash was less frequent in anti-PD-1 (11.0%) and PD-L1 monotherapy (7.9%) [17,18,19,20,21,22,23,24,25,26,27,28,29,30,31,32,33,34,35,36,37,38,39,40,41,42,43,44,45,46,47,48,49,50,51,52,53,54,55,56,57,58,59,60,61,62,63,64,65,66,67,68,69,70,71,72,73,74,75,76,77,78,79,80,81,82,83,84,85,86,87,88,89,90,91,92,93,94,95,96,97,98,99,100,101,102,103,104,105,106,107,108,109,110,111,112,113,114,115,116,117,118,119,120,121,122,123,124,125,126,127,128,129,130,131,132,133,134,135,136,137,138,139,140,141,142,143,144,145,146,147,148,149,150,151,152,153,154,155,156,157,158,159,160,161,162,163,164,165,166,167,168,169,170,171,172,173,174,175,176,177,178,179,180,181,182,183,184,185,186,187,188,189,190].

**Figure 3 cancers-16-00340-f003:**
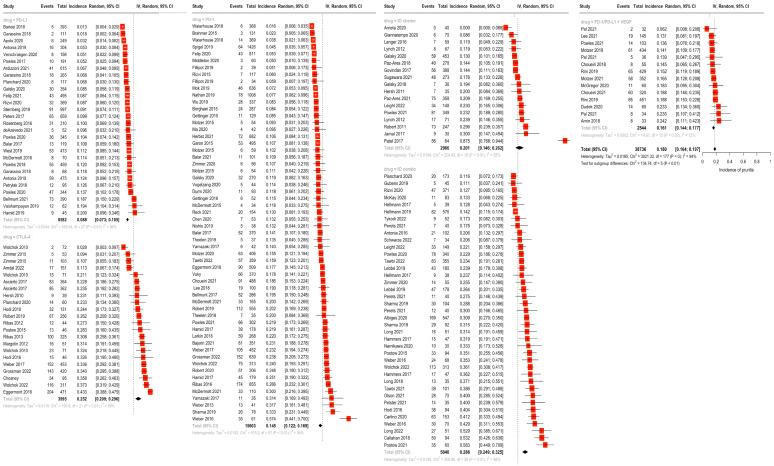
Overall incidence of pruritis by drug class groupings via random-effects modeling of 131 studies reporting pruritis. Highest incidence of pruritis was seen in the IO combination group at 28.6%, followed by CTLA-4 monotherapy (25.2%), IO + chemotherapy (20.1%), IO + antiangiogenic (16.1%), PD-1 monotherapy (14.5%), and PD-L1 monotherapy (8.8%) [17,18,19,20,21,22,23,24,25,26,27,28,29,30,31,32,33,34,35,36,37,38,39,40,41,42,43,44,45,46,47,48,49,50,51,52,53,54,55,56,57,58,59,60,61,62,63,64,65,66,67,68,69,70,71,72,73,74,75,76,77,78,79,80,81,82,83,84,85,86,87,88,89,90,91,92,93,94,95,96,97,98,99,100,101,102,103,104,105,106,107,108,109,110,111,112,113,114,115,116,117,118,119,120,121,122,123,124,125,126,127,128,129,130,131,132,133,134,135,136,137,138,139,140,141,142,143,144,145,146,147,148,149,150,151,152,153,154,155,156,157,158,159,160,161,162,163,164,165,166,167,168,169,170,171,172,173,174,175,176,177,178,179,180,181,182,183,184,185,186,187,188,189,190].

**Figure 4 cancers-16-00340-f004:**
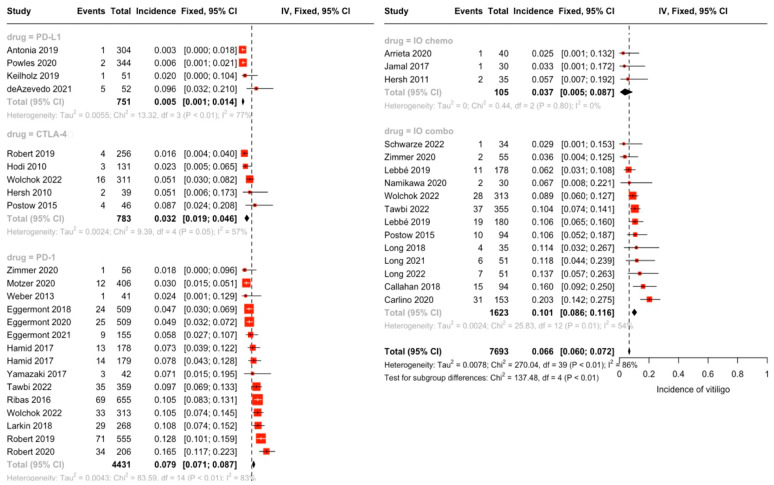
Overall incidence of vitiligo by drug class groupings via fixed-effect modeling of 31 studies reporting vitiligo. Incidence of vitiligo was highest in the IO combination (10.1%) and PD-1 monotherapy (7.9%) groups, followed by the IO + chemotherapy (3.7%), CTLA-4 monotherapy (3.2%), and PD-L1 monotherapy (0.54%) groups. No studies on IO + antiangiogenic combination regimens reported vitiligo [17,18,19,20,21,22,23,24,25,26,27,28,29,30,31,32,33,34,35,36,37,38,39,40,41,42,43,44,45,46,47,48,49,50,51,52,53,54,55,56,57,58,59,60,61,62,63,64,65,66,67,68,69,70,71,72,73,74,75,76,77,78,79,80,81,82,83,84,85,86,87,88,89,90,91,92,93,94,95,96,97,98,99,100,101,102,103,104,105,106,107,108,109,110,111,112,113,114,115,116,117,118,119,120,121,122,123,124,125,126,127,128,129,130,131,132,133,134,135,136,137,138,139,140,141,142,143,144,145,146,147,148,149,150,151,152,153,154,155,156,157,158,159,160,161,162,163,164,165,166,167,168,169,170,171,172,173,174,175,176,177,178,179,180,181,182,183,184,185,186,187,188,189,190].

**Figure 5 cancers-16-00340-f005:**
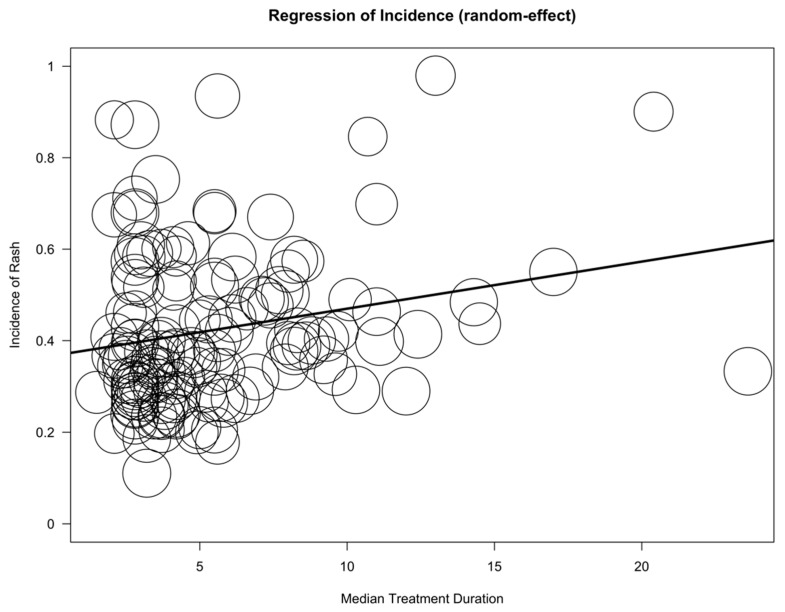
Bubble plot demonstrating the estimated regression slope for incidence of rash and duration of treatment (median months). Rash and median treatment duration were positively correlated, with an expected rise in incidence of 0.01% per month of treatment duration (*p* = 0.0116).

**Figure 6 cancers-16-00340-f006:**
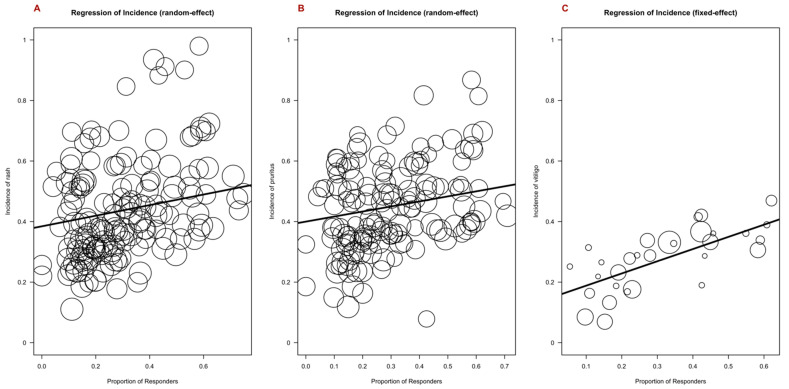
Bubble plot demonstrating the estimated regression slope for incidence of rash (**A**), pruritis (**B**), and vitiligo (**C**), as well as response rate. Response rate was positively correlated with the incidence of pruritis, vitiligo, and rash, with an expected rise in incidence of 0.17% (*p* = 0.0238) for pruritis, 0.40% (*p* = 0.0010) for vitiligo, and 0.18% (*p* = 0.0413) for rash per percentage increase in response rate.

**Figure 7 cancers-16-00340-f007:**
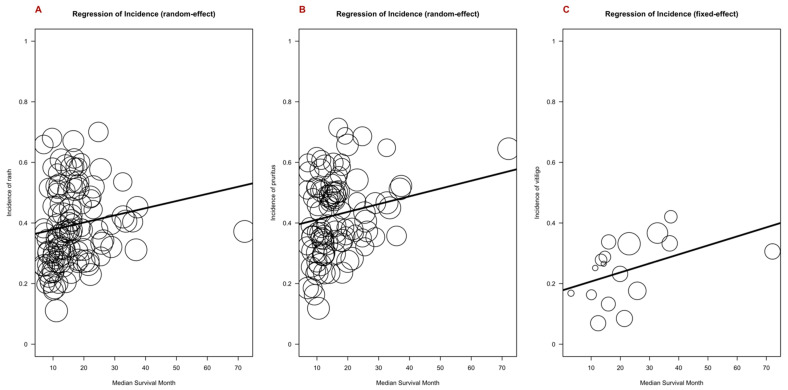
Bubble plot demonstrating the estimated regression slope for incidence of rash (**A**), pruritis (**B**), and vitiligo (**C**), as well as overall survival. Overall survival was positively correlated with the incidence of pruritis, vitiligo, and rash, and this association was significant for vitiligo (*p* = 0.0483), with an expected rise in incidence of vitiligo of 0.003% per additional month of overall survival.

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
