# Peer review of "Incidence of Cutaneous Immune-Related Adverse Events and Outcomes in Immune Checkpoint Inhibitor-Containing Regimens: A Systematic Review and Meta-Analysis"

_cancers, 2024, doi:10.3390/cancers16020340_

Round 1

Reviewer 1 Report

Comments and Suggestions for Authors

1) Abstract. Our analysis provides benchmark incidence rates for cirAES and links cirAEs with favorable treatment outcomes at a study level across diverse solid tumors and multiple ICI regimens. Abstract might be beneficial to include a sentence in the abstract that briefly summarizes the key findings of the study. This can provide readers with a quick overview of the research. 

2) 1. Introduction 34 Immune checkpoint inhibitors (ICIs) have proven to be effective therapies for a vari- 35 ety of cancer types. By targeting immune checkpoints with anti-PD-1, anti-PD-L1, anti- 36 CTLA4, and more recently, anti-LAG-3 agents, ICIs disinhibit the immune system to un- 37 leash anti-tumor immune responses. However, clinically significant off-target effects due 38 to the generation of autoreactive T cells, termed immune-related adverse events (irAEs), 39 are common and can affect virtually any organ. Cutaneous immune-related adverse 40 events (cirAES) are among the most frequently encountered irAEs, occurring at rates of 41 30-40% with anti-PD-1 monotherapy, with potentially higher incidence and severity with 42 combination regimens.1,2. Please add some recent references to support these sentences.

3) Further, the impact of cirAE incidence on 54 survival, response, and duration of therapy has not been robustly examined across ICI- 55 based regimens. In this meta-analysis, we examine the incidence and prognostic associa- 56 tions at a study level of cirAEs across solid tumors and multiple ICI regimens including 57 monotherapy and combination with anti-angiogenic agents, chemotherapy, or other ICIs. Please, improve the description of study aim and underline the novelty of the study.

4) Figure 2-4. Overall incidence of rash by drug class groupings. Please improve the quality of figures.

5) 4. Discussion In our analysis, we provide the most comprehensive meta-analysis of cirAEs by type, therapy regimen, tumor, and correlate cirAEs with outcomes across clinical trials. .... Please improve the description of the most important results of the study.

6) Discussion. The discussion section needs to be improved. It could be interesting to record the aim of the study. It is necessary to be more concise in the presentation of the facts, clarifying the results obtained and comparing them with previous or similar studies. However, it is interesting to answer the questions that arise from these results, backed up by published literature.

6) Our analysis validates the associations of cirAEs with clinical outcomes at a study 294 level across a range of solid tumors and ICI monotherapy and combination regimens and 295 provides benchmark incidence rates across these regimens. As the use of ICIs in a variety 296 of therapeutic combinations and tumors increases, future studies capturing more detail 297 regarding specific cirAEs and patient-level data may allow for more specific prognostic 298 associations to be drawn and inform the care of individual patients. Please, add a separate paragraph regarding the conclusions.

Comments on the Quality of English Language

Minor changes of English language are required

Reviewer 2 Report

Comments and Suggestions for Authors

General comment

The manuscript entitled “Incidence of cutaneous immune-related adverse events and outcomes in immune checkpoint inhibitor-containing regimens: A meta-analysis” by Curkovic et al., presents a comprehensive meta-analysis of immune-related adverse events (cirAEs) in cancer immunotherapy. The study's strength lies in its thorough exploration of cirAEs across different factors, such as cancer types, drug classes, and doses, and its investigation into their associations with treatment outcomes. The organization of the manuscript is clear, facilitating easy comprehension. The manuscript is commended for its extensive coverage of factors affecting cirAEs, contributing to a holistic understanding of their occurrence in cancer immunotherapy. Nevertheless, few issues have to be addressed:

-          The introduction could be extended in order to provide a larger background for the study. To this regard, also see: 10.1016/j.clgc.2023.05.017 and  doi: 10.3390/clinpract13010003

-          The manuscript acknowledges the variability in reporting cirAEs across studies. A more detailed discussion on how this variability may impact the accuracy and reliability of the meta-analysis would enhance transparency.

-          While the study provides insights at the study level, the absence of individual patient-level data limits the ability to draw more specific prognostic associations. A discussion on the implications of this limitation would be beneficial.

-          The manuscript mentions the assumption of linearity in meta-regression. A discussion on the potential implications of this assumption not holding in all cases would enhance the manuscript's analytical transparency.

-          The study highlights associations between cirAEs and treatment outcomes. Including a discussion on the potential clinical implications of these findings for patient management would strengthen the conclusion section.

-          The quality of the figures should be definitely improved.

Comments on the Quality of English Language

Minor language checks

Reviewer 3 Report

Comments and Suggestions for Authors

Dear Authors,

I read your manuscript concerning the incidence of cutaneous immune-related adverse events and outcomes in immune checkpoint inhibitor-containing regimens. The paper is clear, easy to read and reports an exhaustive literature research. I have no major concerns regarding the main text.

I suggest reading and citing the following paper to improve the introduction:

-        Cosio T, Coniglione F, Flaminio V, Gaziano R, Coletta D, Petruccelli R, Dika E, Bianchi L, Campione E. Pyodermitis during Nivolumab Treatment for Non-Small Cell Lung Cancer: A Case Report and Review of the Literature. Int J Mol Sci. 2023 Feb 26;24(5):4580. doi: 10.3390/ijms24054580. PMID: 36902013; PMCID: PMC10003408.

Comments on the Quality of English Language

Minor editing of English language required

Reviewer 4 Report

Comments and Suggestions for Authors

In this review, Curkovic et al performed a meta-analysis of published trials of anti–programmed death-1/ligand-1 (PD-1/PD-L1) and anti–cytotoxic T lymphocyte antigen-4 (CTLA-4) ICIs alone and in combination with chemotherapy, anti-angiogenic agents, or other ICIs in patients with melanoma, renal cell carcinoma, non-small cell lung cancer, and urothelial carcinoma. Across 174 studies, rash, pruritis, and vitiligo were the most reported cirAEs with incidences of 16.7%, 18.0%, and 6.6%, respectively. Higher incidence across cirAEs was associated with ICI combination regimens and with CTLA-4-containing regimens, particularly with higher doses of ipilimumab, as compared to PD-1/L1 monotherapies. Outcome measures including response rate and progression-free survival positively correlated with incidence of cirAEs. Response rate and incidence of pruritis, vitiligo, and rash was associated with an expected rise in incidence of 0.17% (p=0.0238), 0.40% (p=0.0010), 0.18% (p=0.0413), respectively, per percent-age increase in the response rate. Overall survival was positively correlated with the incidence of pruritis, vitiligo, and rash; this association was significant for vitiligo (p=0.0483). They conducted a systematic review and meta-analysis from various angles using 174 suitable references from the large amount of literature of 4600 studies/regimens. I have following questions.

major concerns)

1) Figures 2, 3, and 4 are difficult to see, please make them more legible.

2) For each AE, the ease of appearance differs depending on the mechanism of action. Is this due to differences in the mechanism of action? Or are there other factors? I would appreciate a detailed discussion.

minor concerns)

1) In line 28, you wrote "cirAES and links cirAEs with favorable treatment outcomes at a study level across diverse solid tumors and multiple ICI regimens." Is there a mistake in cirAE "s"? Please correct appropriately.

Reviewer 5 Report

Comments and Suggestions for Authors

The authors performed a meta-analysis to investigate incidence of cutaneous immune-related adverse events and outcomes in immune checkpoint inhibitor-containing regimens. They found that overall survival was positively correlated with the incidence of pruritis, vitiligo, and rash, suggesting incidence of cirAES is favorable for treatment outcomes. The finding is interesting and has clinical significance.  I have some suggestions.

1. This is a systematic review and meta-analysis. Did the authors register their protocol on any website such as PROSPERO? If so, please describe it and provide registration number in the manuscript. If no, please explain why the protocol was not registered.

2. Resolutions of figures are too low, making it hard to read the figures. Please upload figures with high-resolution.

3. Some figure legends are too simplified. Figure legends of Fig. 2, 3, 4 show summarize the main findings of these meta-analysis results.

Round 2

Reviewer 1 Report

Comments and Suggestions for Authors

The manuscript has been improved I have few comments:

1- Please, check the type of manuscrit, I think it is incorrect: article

Manuscripts submitted to Cancers should neither be published previously nor be under consideration for publication in another journal. The main article types are as follows:

  • Article: Original research manuscripts. The journal considers all original research manuscripts, provided that the work reports scientifically sound experiments and provides a substantial amount of new information, e.g., research articles using only one cell line for the experiments will not be considered for publication (experiments need to be repeated on 1-2 more cell lines); authors should consider in vivo studies using orthotopic or transgenic models to validate gene function; for all Western blot figures, densitometry readings/intensity ratio of each band should be included; the whole Western blot showing all bands and molecular weight markers should be included in the Supplementary Materials; gene silencing experiments should use at least two gene-specific siRNAs, etc.

    Authors should not unnecessarily divide their work into several related manuscripts, although short Communications of preliminary, but significant, results will be considered. Quality and impact of the study will be considered during peer review.

    Full experimental details must be provided so that the results can be reproduced. Cancers requires that authors publish all experimental controls and make full datasets available where possible (see the guidelines on Supplementary Materials and references to unpublished data).

    Articles should have a main text of around 3000 words at minimum and should have more than 30 references. Cancers has no restrictions on the maximum length of research manuscripts, provided that the text is concise and comprehensive.

2) Has the manuscript been officially registered as a meta-analysis/systematic review? 

Author Response

Thank you very much for taking the time to review this manuscript and its revisions. Please find the detailed responses below. We have made several small changes to the manuscript as described and highlighted in track changes in the re-submitted files, in addition to changes from prior rounds of editing.  

Comments 1: 1- Please, check the type of manuscript, I think it is incorrect: article

Response 1: Thank you. Based on your feedback, we have requested the publication type be changed to “systematic review” and have subsequently updated the methods section with a statement requested by the editor to allow for this change. Page 3, line 114-116. We have also updated the title to include “systematic review” and reflect this.

Comments 2: Has the manuscript been officially registered as a meta-analysis/systematic review? 

Response 2: Thank you for raising this question. We unfortunately did not register the protocol in a website such as PROSPERO, as we were unaware of this platform at the time of our protocol design and execution. We are unable to register the protocol following completion of the study. We have included a PRISMA flow diagram in the figures, which was generated via the Covidence systematic review software during article review that reflects the protocol used and described in the methods section. Additionally, the PRISMA checklist was used during drafting of the manuscript. We have updated the manuscript to include a statement regarding this, as requested by the editorial office. The methods now read “The systematic review followed the recommendations of the Preferred Reporting Items for Systematic Reviews and Meta-Analyses (PRISMA). The protocol has not been registered.”. Page 3, line 114-116.

Reviewer 2 Report

Comments and Suggestions for Authors

The authors improved the manuscript accordingly to previous suggestions.

Comments on the Quality of English Language

 None

Author Response

Thank you very much for taking the time to review this manuscript and its revisions. We are pleased to receive your feedback that the manuscript is much improved and appreciate your feedback throughout the revision process.

Reviewer 5 Report

Comments and Suggestions for Authors

The authors have responded to my previous comments and revised the manuscript. There is, however, an issue that should be addressed. In the medical research field, it is conventional is to register the protocol before performing/ published the meta-analysis, somehow like clinical trials are required to register protocol. As the authors didn't register their protocol for their meta-analysis paper, there might an issue of acceptabiliy of publication. Some medical journals don't accept meta-analysis that didn't pre-register their protocol. The authors will need clearly indicate why they didn't register protocol before performing the study.

Author Response

Thank you very much for taking the time to review this manuscript and its revisions. Please find the detailed responses below. We have made small changes to the manuscript as described and highlighted in track changes in the re-submitted files, in addition to changes from prior rounds of editing.  

Comments 1: The authors have responded to my previous comments and revised the manuscript. There is, however, an issue that should be addressed. In the medical research field, it is conventional is to register the protocol before performing/ published the meta-analysis, somehow like clinical trials are required to register protocol. As the authors didn't register their protocol for their meta-analysis paper, there might an issue of acceptabiliy of publication. Some medical journals don't accept meta-analysis that didn't pre-register their protocol. The authors will need clearly indicate why they didn't register protocol before performing the study.

Response 1: Thank you for raising this concern. We unfortunately did not register the protocol in a website such as PROSPERO, as we were unaware of this platform at the time of our protocol design and execution. We are unable to register the protocol following completion of the study. We have included a PRISMA flow diagram in the figures, which was generated via the Covidence systematic review software during article review that reflects the protocol used and described in the methods section. Additionally, the PRISMA checklist was used during drafting of the manuscript. We recognize that the absence of protocol registration may hinder acceptability of publication and hope that transparency regarding our methods can effectively communicate the protocol used despite this. We have updated the manuscript to include a statement regarding not registering the protocol, as requested by the editorial office. The methods now read “The systematic review followed the recommendations of the Preferred Reporting Items for Systematic Reviews and Meta-Analyses (PRISMA). The protocol has not been registered.” Page 3, line 114-116.